# Exploring Peripheral Blood-Derived Extracellular Vesicles as Biomarkers: Implications for Chronic Chagas Disease with Viral Infection or Transplantation

**DOI:** 10.3390/microorganisms12010116

**Published:** 2024-01-05

**Authors:** Rafael Pedro Madeira, Paula Meneghetti, Nicholy Lozano, Gislene M. Namiyama, Vera Lucia Pereira-Chioccola, Ana Claudia Torrecilhas

**Affiliations:** 1Disciplina de Infectologia, Departamento de Medicina, Escola Paulista de Medicina, Universidade Federal de São Paulo, São Paulo 04023-900, Brazil; rafaelpedromadeira@gmail.com (R.P.M.); nicholycflozano@gmail.com (N.L.); 2Departamento de Ciências Farmacêuticas, Instituto de Ciências Ambientais, Químicas e Farmacêuticas, Universidade Federal de São Paulo, Diadema 09913-030, Brazil; pmeneghetti@unifesp.br; 3Departamento de Microbiologia, Imunologia e Parasitologia, Escola Paulista de Medicina, Universidade Federal de São Paulo, São Paulo 04023-900, Brazil; 4Electron Microscopy Laboratory, Adolfo Lutz Institute, São Paulo 01246-900, Brazil; ginamiyama@yahoo.com.br; 5Laboratório de Biologia Molecular de Fungos e Parasitas, Centro de Parasitologia e Micologia, Instituto Adolfo Lutz, São Paulo 01246-000, Brazil

**Keywords:** *Trypanosoma cruzi*, extracellular vesicles, chronic Chagas disease patients, immunosuppression, infection reactivation, transplant, viral infection

## Abstract

Extracellular vesicles (EVs) are lipid bilayer envelopes that encapsulate cell-specific cargo, rendering them promising biomarkers for diverse diseases. Chagas disease, caused by the parasite *Trypanosoma cruzi*, poses a significant global health burden, transcending its initial epicenter in Latin America to affect individuals in Europe, Asia, and North America. In this study, we aimed to characterize circulating EVs derived from patients with chronic Chagas disease (CCD) experiencing a reactivation of acute symptoms. Blood samples collected in EDTA were processed to isolate plasma and subsequently subjected to ultracentrifugation for particle isolation and purification. The EVs were characterized using a nanoparticle tracking analysis and enzyme-linked immunosorbent assay (ELISA). Our findings revealed distinctive differences in the size, concentration, and composition of EVs between immunosuppressed patients and those with CCD. Importantly, these EVs play a critical role in the pathophysiology of Chagas disease and demonstrate significant potential as biomarkers in the chronic phase of the disease. Overall, our findings support the potential utility of the CL-ELISA assay as a specific sensitive tool for detecting circulating EVs in chronic Chagasic patients, particularly those with recurrent infection following an immunosuppressive treatment or with concurrent HIV and Chagas disease. Further investigations are warranted to identify and validate the specific antigens or biomarkers responsible for the observed reactivity in these patient groups, which may have implications for diagnosis, the monitoring of treatment, and prognosis.

## 1. Introduction

Chagas disease (CD) is a neglected tropical infection caused by *Trypanosoma cruzi*, affecting millions of people worldwide. CD, primarily transmitted through the bite of infected triatomine bugs, poses a significant public health challenge in various regions. Despite its prevalence, CD often goes undetected and untreated, leading to chronic health issues and, in severe cases, fatalities. This parasitic infection exhibits diverse clinical manifestations, ranging from acute symptoms, such as fever and swelling at the infection site, to chronic complications affecting vital organs like the heart and digestive system.

The chronic phase of CD can remain asymptomatic for years, making its early detection and intervention crucial for preventing long-term health consequences. In addition, CD affects vulnerable and disadvantaged populations significantly, exacerbating existing health disparities. Limited access to healthcare facilities, inadequate funds for disease surveillance, and a lack of knowledge all add to the difficulties in combating this neglected tropical disease. The control and management of Chagas disease necessitate a multifaceted strategy that covers not just medical problems but also social, economic, and environmental variables that contribute to the illness’s persistence [1,2,3,4,5]. While initially concentrated in Latin America, the disease has spread to non-endemic continents via the immigration of infected individuals [2,3,6,7,8,9,10,11].

Transmission can occur via contact with infected triatomine feces, congenital transmission, blood transfusion, laboratory accidents, or oral infection [12,13,14,15]. Following the acute phase, which typically lasts around two weeks, patients enter the chronic phase, where approximately 30–40% of individuals develop clinical manifestations such as digestive, cardiac, and cerebral alterations [2,16,17]. The clinical manifestations of Chagas disease are divided into two phases: acute and chronic [18]. In the acute phase, which begins 6 to 10 days after infection and lasts from 4 to 8 weeks depending on the parasite strain [19], some patients might exhibit clinical signs such as Chagoma and/or Romaña’s sign, and sometimes even arrhythmia, but less than 5% of patients exhibit more severe signs and symptoms [18]. Following the acute phase, 60–70% of patients develop an indeterminate chronic disease that is characterized by positive serology but is clinically asymptomatic [20]. A total of 10–30 years after infection, 30–40% of patients develop symptoms that include severe cardiac complications such as left ventricular dilation and dysfunction, ventricular arrhythmia, congestive heart failure, and acute myocardial infarction [18]. 

Robello’s research group has made important advances in understanding *T. cruzi* biology, host–parasite interaction and development methods, and new technology to treat patients with CD. Their investigation of *T. cruzi* and the factors that lead to CD makes use of a variety of techniques and cutting-edge instruments. These include the first use of RNA-seq in transcriptome research, long-read sequencing to expand the *T. cruzi* genome, chemical proteomics using immobilized benznidazole, and the construction of a genome-wide chromatin interaction map. When these approaches are integrated, they shed light on the critical aspects of parasite–host interactions and parasite biology. Each addition adds to our understanding of the transcriptome profiles, genomic architecture, proteomic targets, and chromatin landscape organization of *T. cruzi* [21,22,23,24,25,26,27,28].

In the context of host–pathogen interactions, extensive research has been conducted to elucidate the molecular mechanisms involved [29,30,31,32]. *T. cruzi* molecules have been identified and characterized by highlighting their ability to activate macrophages and induce the production of pro-inflammatory chemokines and cytokines [33,34,35,36,37,38]. Molecules anchored by glycosylphosphatidylinositol (GPI) and trans-sialidase (TS) expressed by *T. cruzi* play crucial roles in triggering immune responses and stimulating the production of inflammatory mediators [34,37,39,40]. EVs have emerged as key players in intercellular communication and modulation [41,42,43,44,45]. These membrane-derived vesicles can be classified into microvesicles, exosomes, and apoptotic bodies, each with distinct biogeneses, release pathways, sizes, contents, and functions [41,42,43,46,47]. Previous studies have demonstrated that *T. cruzi* trypomastigotes release EVs containing glycoproteins, including TS, gp85, and mucin, which play essential roles in modulating the host innate immune response [31,48,49,50]. In Chagas disease, these glycoproteins serve as targets for lytic anti-αGal antibodies, which contribute to the control of parasitic infection [51]. 

Our recent investigations have shown decreased concentrations of EVs released by peripheral blood mononuclear cells in chronic Chagas disease patients compared to healthy individuals [52]. This finding supports observations from circulating EVs in Chagas disease patients [52,53].

The method employed for isolating EVs is essential, as it determines the test results. A typical method consists of centrifuging plasma at 100,000× *g* for 1 h, with pellets resuspended in filtered PBS. Another method is to isolate EVs using three rounds of centrifugation at 15,000× *g* for 15 min. Size exclusion chromatography (SEC) is used to isolate EVs from plasma for proteome investigation, giving excellent purity by eliminating soluble plasma proteins while preserving important EV properties. Frozen plasma samples are thawed, centrifuged, and put onto self-packed or commercially available Sepharose CL-4B columns for elution in PBS, giving fractions for analysis. The approach used is determined by the downstream analysis requirements, considering the benefits of each [54,55]. Given the importance of EVs in *T. cruzi* infection and their potential implications, we aim to characterize the EVs present in the plasma of patients experiencing the reactivation of chronic Chagas disease. In addition to our investigation, we will take time to examine the complex molecular expressions of Gal-mucin and trans-sialidase (TS). This research aims to improve our understanding of the complex relationship between EVs and *Trypanosoma cruzi,* the causative agent of Chagas disease. Alpha-Gal-mucin, a glycoprotein, is important in host–parasite interactions, maintaining *T. cruzi* recognition and adhesion to host cells [40]. Trans-sialidase, an enzymatic virulence factor secreted by the parasite, regulates the host immune response and enhances invasion at the same time. We hope to unravel the underlying mechanisms driving the dynamics of EVs and *T. cruzi* in the setting of Chagas disease by examining the slight shifts on these molecules’ expression. This extensive research will not only help us understand the complex molecular events that take place during infection, but it will also provide crucial insight into potential therapies for reducing the impact of Chagas disease. We aim to understand the complexity of host–parasite interactions and open a way for new approaches in the treatment and therapy of this tropical disease by carefully investigating Gal-mucin and TS expressions in the patients.

## 2. Material and Methods

### 2.1. Ethics Statement

This research was approved by the Research Ethics Committee of the Institutional Review Board of Universidade Federal de Sao Paulo (CEP/UNIFESP-CAAE: 70749317.2.0000.5505). Samples were obtained from Chagasic patients and healthy individuals (controls) who willingly provided written informed consent. Our study enrolled subjects with a confirmed positive serology for Chagas disease, without any limitations on gender or age. Control individuals had a negative serology for Chagas disease.

### 2.2. Healthy Controls, Patients, Clinical Samples, and Laboratory Diagnosis

The control group comprised 30 health individuals who tested negative for Chagas disease and had no known medical conditions at the time of sample collection for the isolation of EVs from plasma. The samples consisted of 44 plasma samples collected from patients diagnosed with Chagas disease over a period of 12 months, specifically from December 2018 to November 2019. The patients included in the study were diagnosed with Chagas disease both clinically and with laboratory tests and were categorized into different groups based on their clinical characteristics. Group I consisted of patients who experienced a recurrence of infection due to immunosuppressive treatment after undergoing transplantation or cancer therapy. These patients were selected to investigate the presence of Chagas disease recurrence in the context of immunosuppression. Group II comprised patients with CCD, focusing on individuals with persistent infection and long-term clinical manifestations of the disease. These patients were selected to analyze the presence of Chagas disease markers in patients with established chronic infection. Group III consisted of patients with a co-infection of HIV and Chagas disease. This subgroup was included to assess the impact of HIV co-infection on the clinical presentation and molecular characteristics of Chagas disease. For the molecular diagnosis of Chagas disease, blood samples were collected in tubes containing ethylenediaminetetraacetic acid (EDTA) as an anticoagulant. The samples were then sent to Instituto Adolfo Lutz in São Paulo, Brazil for further analysis. To extract the DNA, the blood samples were centrifuged for 15 min at 1850× *g*, and the resulting blood pellets were utilized. To proceed with DNA extraction, the cells obtained from the blood pellets were digested using 20 µg of proteinase K in AL buffer (Qiagen). Subsequently, DNA extraction was performed using the QIAamp DNA Mini Kit (Qiagen) according to the manufacturer’s instructions. The purity and concentration of the extracted DNA were assessed using a NanoDrop ND1000 spectrophotometer. To detect the presence of T. cruzi DNA, a real-time polymerase chain reaction (qPCR) was performed. A specific primer set, *T. cruzi* 32/148, was used for amplification. This primer set consisted of the forward primer 32 (5′ TTTGGGAGGGGCGTTCA-3′) and the reverse primer 148 (5′ ATATTACAC-CAACCCCAATCGAA-3′). The MGB TaqMan probe 71 (5′ CATCTCACCCGTACATT3′) was labeled with FAM and NFQ at the 5′ and 3′ ends, respectively, and used for detection. All 44 patients included in the study tested positive for *T. cruzi* via qPCR, indicating the presence of the parasite’s DNA in their blood samples. This confirmed the diagnosis of Chagas disease in the patient cohort and provided a basis for the further analysis of the clinical and molecular characteristics associated with the different patient groups. Overall, this comprehensive evaluation of plasma samples from patients diagnosed with Chagas disease revealed the persistence of *T. cruzi* DNA in all cases. The categorization of patients into distinct groups allowed for the examination of specific aspects related to disease recurrence, chronic infection, and co-infection with HIV. These findings contribute to a deeper understanding of the molecular features and clinical implications of Chagas disease in different patient populations. 

### 2.3. Isolation and Characterization of EVs from Human Plasma Samples from Healthy Controls and Patients

The protocol employed in this study was based on the methodologies described by Madeira et al. [52,53]. To perform the ultra-centrifugation of plasma samples, a fixed angle rotor, specifically the Thermo Scientific™ T-8100 Fixed Angle Rotor (Waltham, MA, USA), was utilized. The ultra-centrifugation process was carried out using the Thermo Scientific™ Sorvall™ WX100 Ultra Centrifuge, with a centrifugal force of 100,000× *g*, for a duration of 1 h. To initiate the ultra-centrifugation process, 1 mL of plasma sample was carefully transferred to a suitable centrifuge tube. It is important to note that proper handling and preparation of the samples are critical to ensure reliable and reproducible results. The use of sterile techniques, such as working in a laminar flow hood and wearing appropriate personal protective equipment, is recommended to minimize the risk of sample contamination. Once the plasma samples were loaded into the centrifuge tubes, the tubes were securely sealed to prevent any leakage or loss of sample during the centrifugation process. It is essential to select centrifuge tubes that are compatible with the rotor being used to ensure optimal performance and safety. During the ultra-centrifugation process, the centrifuge operated at a force of 100,000× *g*, subjecting the plasma samples to a strong gravitational field [52,53]. 

In the next step, we performed size-exclusion chromatography (SEC). Initially, the filtered samples (0.1 mL) were adjusted to a volume of 1 mL by using a 100 mM ammonium acetate solution at pH 6.5. These EV samples were subsequently loaded into a Sepharose CL-4B column (1 × 40 cm, Cytiva, Marlborough, MA, USA), which had been pre-equilibrated with 100 mM ammonium acetate at pH 6.5. The column was then eluted with the same buffer at a flow rate of 0.2 mL/min. In total, 30 fractions were obtained (1 mL each fraction). All fractions were screened using CL-ELISA, following the methods outlined in a prior publication by our group [53,56].

In parallel, we performed the Dot Blot. The nitrocellulose membrane (NM) was employed, with a grid drawn on it to demarcate the blotting region. Subsequently, 5 μL (10^3^ EVs/spot) of each EV sample was spotted at the center of the grid, and the membrane was left to air-dry at room temperature (RT). Following this, it was blocked overnight with 5% nonfat milk in Tris-buffered saline (TBS). The NM was then exposed to the following antibodies: (i) rabbit polyclonal anti-human CD9 (Thermo Scientific 11559); (ii) mouse monoclonal anti-human CD63 (Immunostep, Salamanca, Spain, 63PU); (iii) mouse monoclonal anti-human CD86 (Thermo Scientific 80820); and (iv) mouse monoclonal anti-human CD82 (Thermo Scientific 27233). All antibodies were diluted at 1:500 in TBS with 5% nonfat milk. After incubating for 1 h at RT, the NM underwent TBS-Tween washing and was then exposed to anti-rabbit IgG-peroxidase conjugate or anti-mouse Ig-peroxidase (both from Kirkegaard & Perry Laboratories, Gaithersburg, MD, USA) at a 1:5000 dilution in TBS with 5% nonfat milk for 1 h. After five additional washes in TBS-Tween, the peroxidase signals were detected by introducing a chemiluminescent (ECL) solution (A-38555, Thermo Fisher Scientific) and incubated in the dark for 15 min, with images acquired using an Odyssey C imaging system (Li-Cor) [56]. 

For Transmission Electron Microscopy (TEM), EVs isolated from both HC and patients’ samples, obtained via size-exclusion chromatography, as previously detailed [53], were examined to assess their sizes and structural integrity. For Transmission Electron Microscopy (TEM) Negative Staining Protocol, EVs (10^9^ particles/200 μL isolated from patients and health controls) were initially preserved in a solution of 2% paraformaldehyde in phosphate-buffered saline (PBS) with a volume ratio of 1:1 for a duration of 1 h. Subsequently, a single droplet of the EV suspension was applied onto an electron microscopy (EM) grid and subjected to negative staining using a 2% potassium phosphotungstate solution adjusted to pH 6.8, following established procedures. The prepared grids were then examined using a JEOL Transmission Electron Microscope, Akishima, Japan (model JEM1011) operating at 80 kV. Imaging was conducted with a Gatan 785 ES1000W Erlangshen camera (Pleasanton, CA, USA) [57].

### 2.4. Nanoparticle Tracking Analysis (NTA)

The isolated EVs were appropriately diluted and subjected to NTA using a NanoSight NS300 instrument manufactured by Malvern Panalytical Ltd. (Malvern, UK). The NTA analysis was performed with a specific set of parameters to ensure accurate and reliable measurements. The NanoSight NS300 instrument was equipped with a sCMOS camera and operated at a wavelength of 532 nm. The camera level was set to auto, allowing for optimal adjustment of the camera’s sensitivity to capture EVs with high resolution. The detection threshold was set to 10, ensuring that only particles above this threshold were considered for analysis. Manual focus adjustment was performed to achieve clear imaging of the EVs during the tracking process. For the NTA analysis, a volume of 500 µL of the isolated EV samples was manually injected into the laser chamber of the NanoSight NS300 instrument. The injection was carefully carried out to prevent any introduction of air bubbles or sample loss. Readings were taken in triplicate for each sample, with each reading lasting 30 s at a frame rate of 25 frames per second. This allowed for the tracking and measurement of EVs in real-time, capturing their Brownian motion.

### 2.5. Chemiluminescent Enzyme-Linked Immunosorbent Assay (CL-ELISA) for the Detection of α-Gal, TS, and Parasite Membrane Molecules on EVs from Human Plasma from Healthy Individuals (Controls) and Patients

EVs (10^3^ particles/well) were analyzed using a CL-ELISA to detect specific antigens related to Chagas disease. The following primary antibodies were used: anti-α-Gal (human polyclonal antibody), anti-TS (mouse monoclonal antibody), and anti-*T. cruzi* membrane antibodies (rabbit polyclonal antibody). To perform the CL-ELISA, 96-well plates were coated with the isolated EVs and incubated overnight to ensure proper immobilization of the antigens. The plates were then washed three times with phosphate-buffered saline (PBS) to remove any unbound material. Subsequently, the wells were blocked for 1 h using 5% nonfat dried milk to minimize non-specific binding and reduce background noise. After the blocking step, the plates were washed again with PBS containing 0.05% Tween 20, a detergent used to enhance the efficiency of washing and reduce non-specific interactions. Following the washes, the plates were incubated for 1 h with the primary antibodies at appropriate dilutions. The anti-α-Gal antibody was diluted at 1:800, the anti-*T. cruzi* membrane antibody was diluted at 1:1000, and the anti-TS antibody was diluted at 1:1000. These primary anti-bodies specifically target the α-Gal epitope, *T. cruzi* membrane proteins, and TS antigens, respectively, enabling the detection of relevant markers associated with Chagas disease. In parallel, after washing with PBS containing 0.05% Tween 20, the plates were incubated with horseradish peroxidase (HRP)-conjugated secondary antibodies for 1 h. The secondary antibodies, including anti-rabbit, anti-mouse, and anti-human antibodies, were diluted at 1:1000 and targeted the respective primary antibodies used in the assay. This step further amplified the signal generated by the specific antigen–antibody interactions. The plates were washed with PBS containing 0.05% Tween 20 to remove any unbound secondary anti-bodies. A chemiluminescent substrate was added to the wells, and the plates were read at approximately 425 nm using an appropriate detection instrument. The chemiluminescent reaction generated light in proportion to the presence of the target antigens bound to the antibodies immobilized on the plate [56].

### 2.6. Data Analysis

Data sets were compared using unpaired *t*-test with Welch’s correction, and one-way ANOVA with Tukey multiple comparisons test, using GraphPad Prism 10 software. Differences in concentration and modal size of the EVs were analyzed using multiple comparisons and corrected for multiple testing (one-way ANOVA with post hoc Tukey test), assuming a value of *p* < 0.05 to be significant, as described earlier.

## 3. Results

### 3.1. Distribution of Patients with CCD and Controls

The distribution of the cohort of Chagas disease patients according to gender revealed a total of 44 individuals, with 23 (52%) being male and 21 (48%) being female. Regarding age groups, the patients were categorized as follows: 6 (14%) individuals were less than 40 years old, 28 (64%) fell between the ages of 40 and 60 years, and 10 (22%) were over 60 years old. No significant differences were observed in our analysis. We believe that our sample was carefully stratified based on the parasite detection results via PCR, as well as based on age and gender, thereby ensuring the representativeness of our results within the studied population. Every patient included in our study was undergoing a reactivation of infection. Upon further analysis, where patients were categorized according to clinical forms and comorbidities, the subsequent outcomes emerged: Group I, comprising 24 (55%) patients, demonstrated an infection relapse that was primarily caused by immunosuppressive treatments following transplantation or cancer; Group II consisted of 16 (36%) patients diagnosed with chronic Chagas disease; finally, Group III included 4 (9%) patients with a co-infection of HIV and Chagas disease. 

These findings are graphically depicted in Figure 1 and concisely summarized in Appendix A, providing an inclusive perspective of the distribution of chronic Chagas disease reactivation across diverse patient demographics and clinical presentations. This control group was integrated into the study to function as a reference cohort for the purpose of comparing both age and sex demographics with the Chagas patients’ samples. The healthy control (HC) group was meticulously characterized by both gender and age. In this cohort, there were 10 males and 20 females, with a total of 30 control samples. The age distribution encompassed various categories, including individual; HCs aged 20 years old (N = 1); 20–39 years old (N = 27); and 40 to 59 years old (N = 2). This comparative strategy was employed to thoroughly scrutinize any observable discrepancies and to evaluate the potential ramifications of EV markers.

We are confident that there are no lipoproteins binding to *T. cruzi* antigens present in the EVs after the purification of plasma particles from healthy people and patients using ultracentrifugation and size exclusion chromatography (SEC), as we would not detect a positive response from specific antibodies to the parasite’s glycoconjugates (there could be a detection block in the CL-ELISA). Additionally, we only used the fractions of the chromatography that were positive for EV markers, like CD9, CD63, CD82, and CD86. The Dot Blot was then carried out using the same markers’ EVs. The parasite antigens were detected using the CL-ELISA assays with positive findings. Thus, the EV preparation is properly fractionated, purified, enriched, and characterized using SEC and transmission electron microscopy (TEM). According to Figure 2 and Figure 3 and Appendix A, which shows representative samples of extracted and purified EVs from healthy people and patients, we performed a Dot Blot using the primary EV markers. We are confident that there are no lipoproteins in the sample preparation, especially none that could interact with *T. cruzi* antigens. In parallel, we performed the CL-ELISA with EVs isolated from patients and healthy controls using antibodies against the parasite plasma membrane molecules, such as alpha-Gal and TS. Most antigenic epitopes expressed on the surface of the parasite’s plasma membrane are recognized by these antibodies. All antibodies that target the parasite membrane and other parasite antigens recognized EVs isolated from chronic Chagas patients. In CL-ELISA, there is a variation in the intensity of detection in the EV samples from patients (Figure 4). The detection threshold for the EVs isolated from the negative controls was less than 400 URL. As a result, we decided to conduct additional tests, stratifying the groups based on age, gender, and the cause of Chagas disease reactivation in the patients. These protocols follow the MISEV guidelines and the Guidelines for the Purification and Characterization of Extracellular Vesicles of Parasites [44,45]. 

### 3.2. EV Size Did Not Differ Much among Groups, but Clusters of Patients Can Be Observed

The average size of EVs purified from the plasma of patients with Chagas disease was approximately 150 nm, as determined by NTA (Figure 5A–C). Interestingly, no significant differences in the EV size were observed when considering factors such as age, gender, or the clinical manifestation of the disease. However, upon examining the individual values and plotting them together, two distinct subpopulations of EVs emerged based on their sizes: one with diameters of around 200 nm and another with diameters of around 100 nm. This phenomenon was particularly evident when patients were grouped by sex and age. Regarding the concentration of EVs from the plasma of patients with Chagas disease, the mean values ranged from approximately 1 × 10^8^ to 1 × 10^9^ EVs/mL (Figure 5D–F). Although the graphical representation indicated certain trends among the mean concentrations, no statistically significant differences were observed.

### 3.3. Differential Expression of Parasite Molecules Demonstrates Variability across Distinct Age Groups and Immunosuppression Categories

In parallel, we investigated the expression of *T. cruzi* molecules in EVs derived from patients’ plasma. Three specific molecules, the level of α-Gal present in the glycoproteins, like mucin, TS, and total *T. cruzi* membrane proteins, were analyzed. The results are displayed in Figure 6, representing α-Gal expression and detection; Figure 7, which shows TS expression/detection; and Figure 8, which illustrates the expression of total *T. cruzi* membrane proteins. Remarkably, a considerable proportion of patients exhibited increased levels of α-Gal, particularly with advanced age, as depicted in Figure 6. Conversely, TS expression was found to be low in these samples, as observed in Figure 7. This phenomenon of TS expression was also observed when comparing immunosuppressed patients to non-immunosuppressed (indeterminate) Chagas patients (Figure 7). Notably, clusters of patients were evident when analyzing TS expression (Figure 7), like the observations in mean size analysis (Figure 5). Gender did not appear to be a confounding factor on specific molecule expression. In addition, there were no differences among groups for the total *T. cruzi* membrane molecules (Figure 8). In the CL-ELISA assays, the control samples were EVs isolated from healthy donors. Notably, our results demonstrated a lack of reactivity in the EVs isolated from donors, thereby highlighting the assay’s specificity for circulating EVs derived from chronic Chagasic patients. This specificity is particularly relevant in the context of patients experiencing a recurrence of infection due to immunosuppressive treatment following transplantation or cancer. Moreover, statistically significant results were obtained when analyzing patients diagnosed with chronic Chagas disease and those with the co-infection of HIV and Chagas disease. Moreover, upon evaluating the frequency of immunosuppressed patients in relation to parasite molecule expression, it was found that there were no significant differences in the expressions of αGal or total membrane proteins (Figure 9A and Figure 9B, respectively). However, our data revealed a higher frequency of HIV-immunosuppressed patients with elevated expressions of trans-sialidase (Figure 9B).

## 4. Discussion 

In the past decade, the applications of EVs in clinical therapy have witnessed remarkable advancements [31,50,58]. However, despite the growing interest, several challenges persist in the clinical investigation of EVs, particularly regarding their utilization in clinical trials and their application in monitoring the progression of inflammatory and infectious diseases [54,55,59,60]. One major obstacle is the absence of reliable biomarkers for tracking the progression of infectious diseases. Animal models have demonstrated the significant involvement of EVs in the development of heart parasitism and inflammation during infection [49,61]. Taking inspiration from these findings, we aimed to characterize the circulating population of EVs in the peripheral blood of patients diagnosed with chronic Chagas disease (CCD). Chagas disease, which is caused by the parasite *T. cruzi*, is a chronic inflammatory condition affecting millions of individuals worldwide. By studying the EVs in CCD patients, we aimed to gain insights into their compositions, functions, and potential immunomodulatory capacities. To accomplish this, we designed an experimental approach involving both in vivo and in vitro investigations. First, we focused on analyzing the peripheral blood circulating population of EVs in CCD patients. We collected blood samples from a cohort of CCD patients and isolated EVs using established techniques such as ultracentrifugation or size-exclusion chromatography. The isolated EVs were then subjected to detailed characterization, including their size distributions, surface marker expressions, cargo contents, and subpopulation heterogeneity. This analysis aimed to provide a comprehensive understanding of EV profiles in CCD patients, shedding light on their potential roles in disease pathogenesis and progression [52,53,54,55,60]. By characterizing EVs in CCD patients and evaluating their immunomodulatory potential, our study aimed to contribute to the understanding of pathogenesis and immune dynamics in Chagas disease [16,31,62,63,64]. The results obtained from our investigations have the potential to uncover new insights into the roles of EVs in infectious diseases and pave the way for the development of novel diagnostic and therapeutic strategies. Ultimately, by harnessing the knowledge gained from studying EVs, we aim to advance the field of clinical therapeutics and improve patient outcomes in the context of inflammatory and infectious diseases.

An increase or decrease in the concentration of circulating EVs has been observed in various inflammatory and infectious processes [52,53,54,55,60]. For instance, patients with periodontitis exhibit increased EV concentrations in the crevicular fluid, which correlates with clinical inflammatory parameters [65]. In immunocompromised individuals, such as those with B cell lymphoma treated with rituximab, the ability to mount an appropriate immune response against pathogens is impaired, potentially leading to disease relapse, even after achieving a clinical cure [66].

In our study, we observed a trend where female patients had higher concentrations of circulating EVs compared to male patients. This finding could be influenced by factors such as menstrual cycle stage, as women exhibit variable EV concentrations depending on the stage of their menstrual cycle. In addition, these differences tend to diminish with age [67,68]. It is worth noting that most of our patients below the age of 40 were women, which may contribute to the observed differences.

The glycoconjugates present on the parasite membrane play crucial roles in the pathophysiology and diagnosis of Chagas disease. The α-galactose epitopes, located on mucins anchored to the membrane by the glycophosphatidylinositol, are highly immuno-genic and stimulate the production of lytic antibodies in patients with Chagas disease [39,40,51,69,70]. Another important parasite molecule involved in parasite–host interactions is the trans-sialidase enzyme, which plays roles in cell invasion and complement evasion and induces IFN-γ production by peripheral blood mononuclear cells [29,30,31,48,63,71,72]. In our studies with patients with disease reactivation, we detected the presence of both α-galactose and trans-sialidase, as well as other unidentified parasite proteins. 

Interestingly, our results showed a positive correlation with α-galactose and a negative correlation with trans-sialidase when stratifying our patient population by age. This was expected, as changes in immune response occur due to immunosenescence, the gradual decline of immune function with age [73].

The impairment of immune response development is also observed in patients undergoing transplantation, such as liver transplant recipients, who have a higher risk of infection or reactivation of previous cytomegalovirus infections [74]. Furthermore, the in vitro co-infection of astrocytes with *T. cruzi* and HIV leads to changes in the microenvironmental oxidative state, affecting parasite growth and development [75]. 

Our results found heterogeneity in protein expression across our patients, particularly when stratified by immunosuppression, which highlights the importance of variable immune response circumstances in Chagas disease. Such changes are not commonplace in immunosuppressed clinical settings, adding a degree of complexity to the illness dynamics.

During chronic phase reactivation, the complicated interplay between immunological factors and the expression patterns of important parasite molecules, such as alpha-galactose and trans-sialidase, is especially visible. The observed differences suggest a possible link between altered immune responses and dynamic changes in EV content during the chronic phase of Chagas disease. This discovery demands an in-depth analysis of the intricate mechanisms at work, indicating the need for additional research to untangle the complexities of the immune–parasite interplay and its implications for Chagas disease management.

Understanding how immune responses are altered during acute reactivation and the subsequent influence on parasite compounds transported by EVs becomes critical in the context of Chagas disease. This dynamic connection between the host immune system and parasite components complicates illness progression and necessitates a thorough examination into the underlying mechanisms. Furthermore, the impact of immunosuppression in influencing disease progression cannot be overstated. The altered protein expression found in immunocompromised people raises questions regarding how immune modulation affects parasite survival and persistence. Untangling these links is critical for developing effective therapeutic methods for Chagas disease patients, especially in clinical settings, where immunosuppression is widespread. Furthermore, the implications of the α-galactose variable and trans-sialidase expression on EVs extend beyond mere markers of disease activity. These variations may serve as indicators of the host–parasite interaction dynamics and, potentially, as prognostic factors for disease outcomes. Investigating these aspects is crucial for developing targeted interventions and personalized treatment approaches in Chagas disease management.

When we look more into the complex nature of the immune system and its impact on parasite molecules, it becomes evident that these interactions are not isolated occurrences, but rather sophisticated processes that occur over the course of the disease progression. Understanding the time dynamics of these events is critical for devising therapies that address Chagas disease’s changing nature. The observed intricacy in immune response modulation and its link with parasite compounds on EVs emphasizes the importance of long-term research. Following patients over time will allow for a more complete picture of the changing host–parasite dynamics, providing essential insights into the factors determining illness severity, progression, and treatment response.

In conclusion, the complex interaction between immunosuppression, immune response modulation, and parasite molecule expression on EVs emerges as a critical determinant in the chronic phase of Chagas disease. This complication emphasizes the importance of continuing research efforts to identify the underlying mechanisms, providing a route for more targeted and successful Chagas disease management strategies.

## Figures and Tables

**Figure 1 microorganisms-12-00116-f001:**
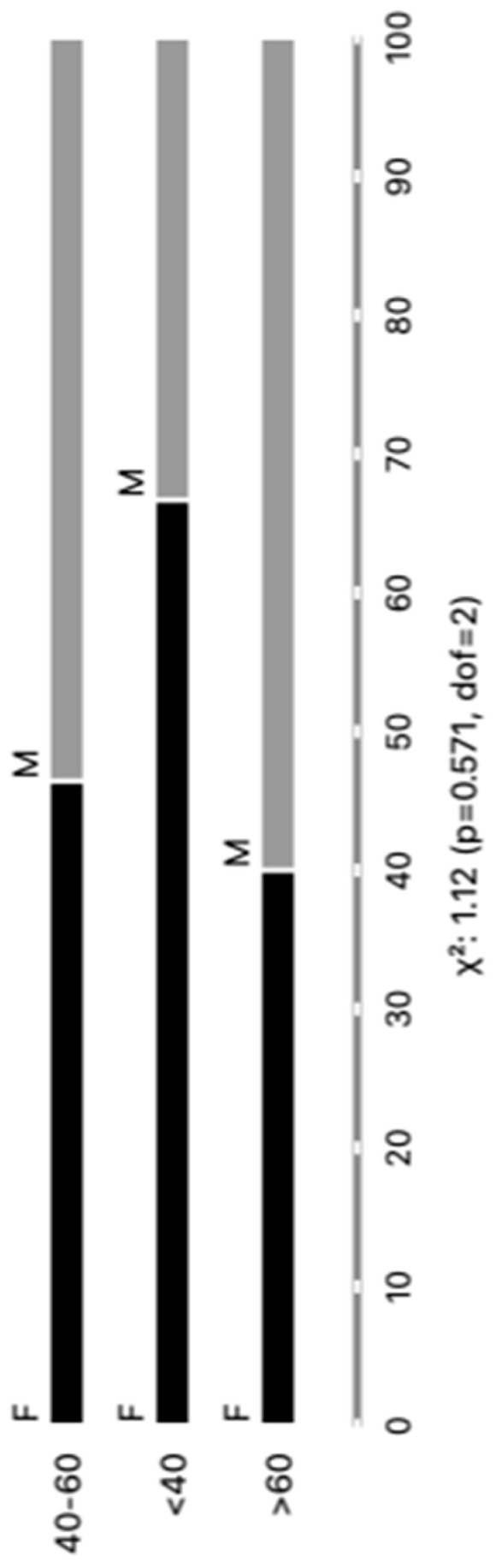
Categorizing individuals with positive PCR results based on age and gender (M = male and F = female).

**Figure 2 microorganisms-12-00116-f002:**
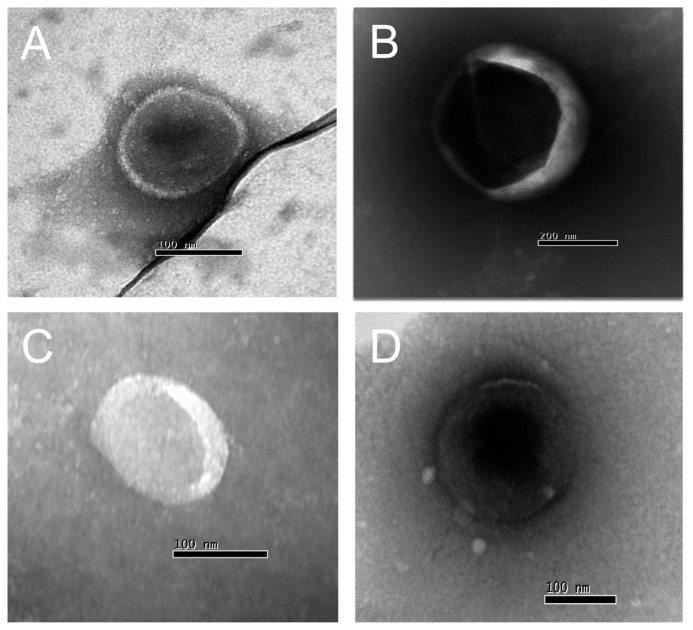
Transmission electron microscopy (TEM) of EVs isolated from chronic Chagasic patients (**A**,**B**) and healthy controls (**C**,**D**) (100–300 nm diameter).

**Figure 3 microorganisms-12-00116-f003:**
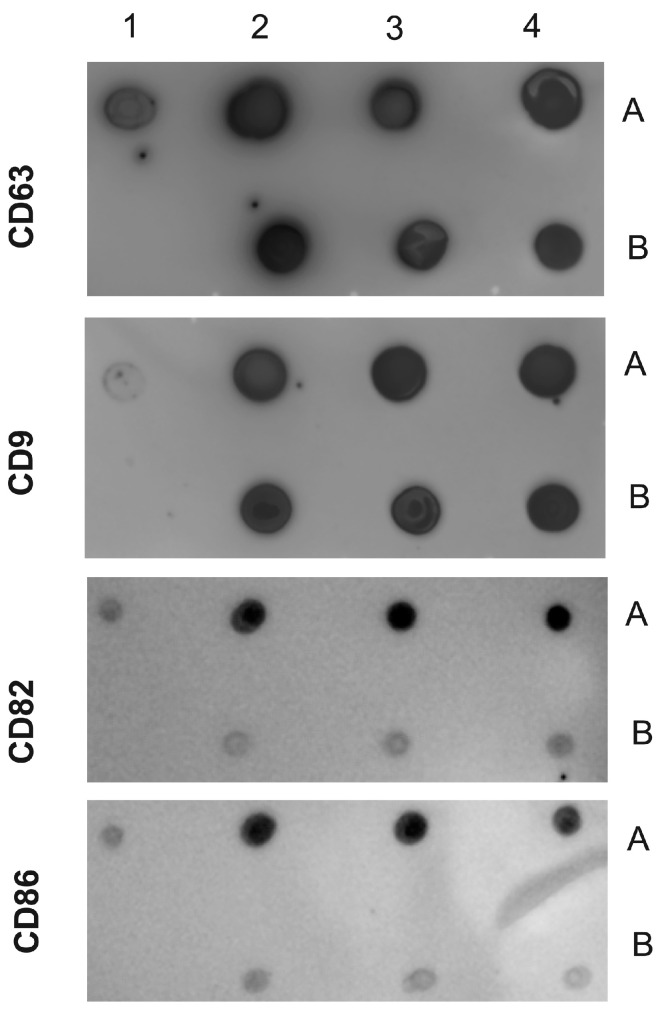
Dot Blot of EVs isolated from chronic Chagasic patients and controls (healthy individuals). The following samples were spotted on the NM antibodies against CD9, CD63, CD82, and CD86. Positive control (1A); negative control (1B). EVs purified from chronic Chagasic patients (2A, 3A, and 4A) and EVs from healthy controls (2B, 3B, and 4B). The MNs were incubated with the indicated antibodies and revealed with the respective peroxidase conjugates.

**Figure 4 microorganisms-12-00116-f004:**
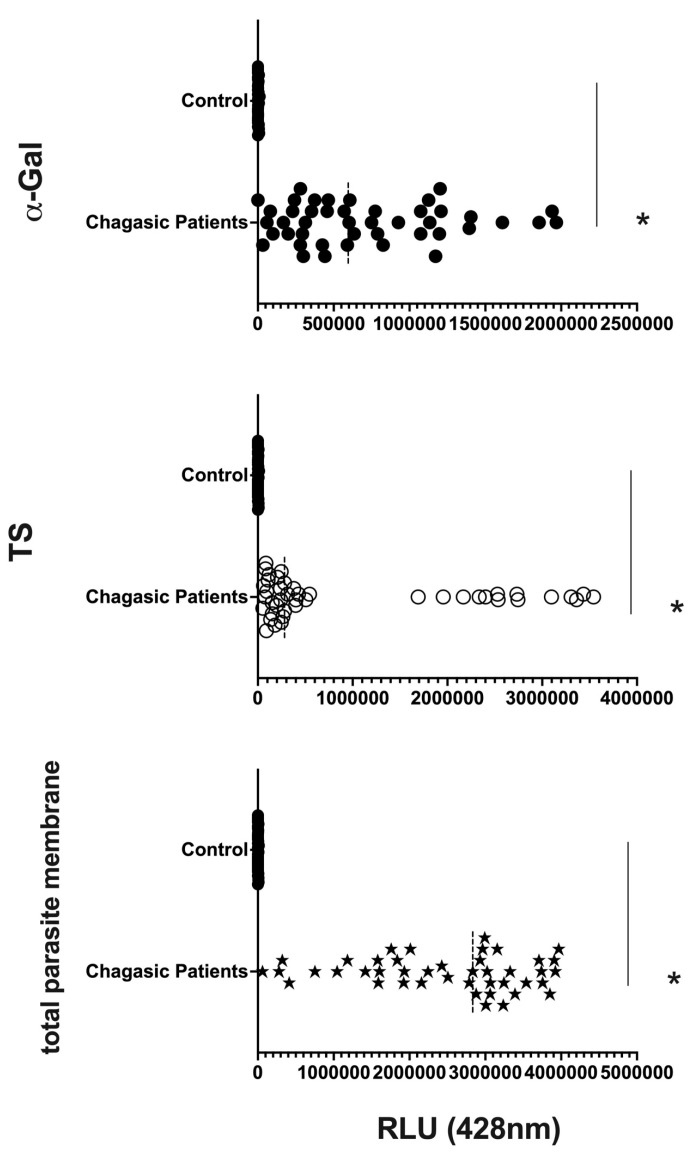
Detection of expression of total membrane, α-Gal, and TS epitopes in all EV samples from chronic Chagasic patients (PCR-positive individuals with CCD) and EVs isolated from healthy controls, measured in Relative Light Units (RLUs). * The calculated *p*-value of the statistical analysis was less than 0.05, signifying the statistical significance of the outcomes.

**Figure 5 microorganisms-12-00116-f005:**
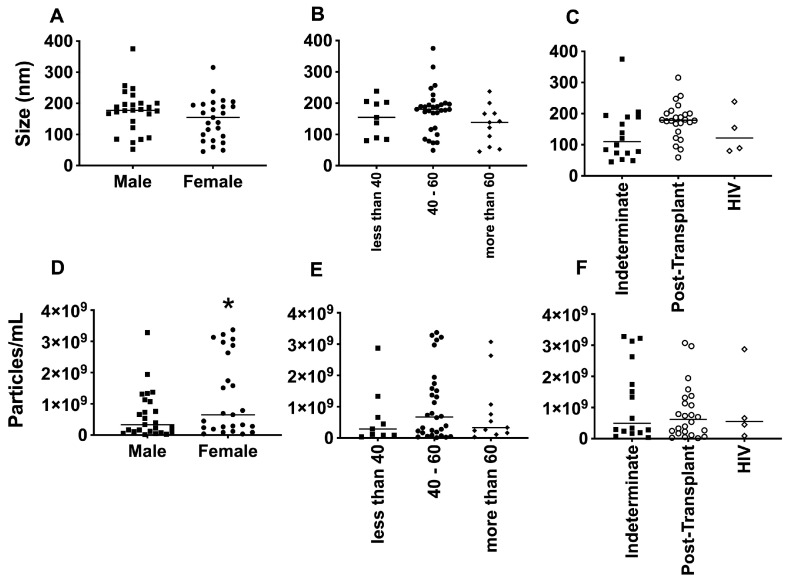
Determination of the average size, in nanometers (**A**–**C**), and average concentration, in particles per milliliter (**D**–**F**), of EVs from PCR-positive individuals with CCD. Categorization based on gender (**A**,**D**), age (**B**,**E**), and immunosuppression status (**C**,**F**). The “Indeterminate” group refers to immuno-competent chronic Chagas disease patients. * The *p*-value obtained from statistical analysis was less than 0.05, indicating that the results were statistically significant.

**Figure 6 microorganisms-12-00116-f006:**
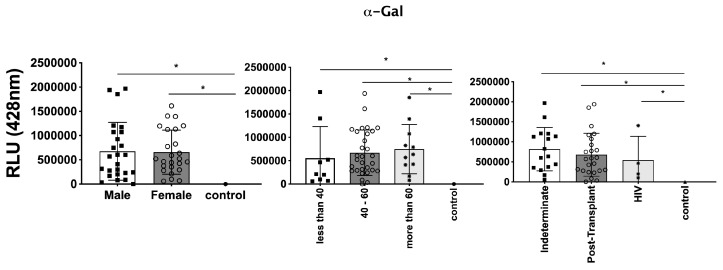
Detection of expression of α-Gal epitopes in EVs from PCR-positive individuals with CCD and control, measured in Relative Light Units (RLUs). Categorization based on gender (**Left**), age (**Centre**), and immunosuppression status (**Right**). The “Indeterminate” group refers to immunocompetent CCD patients. The control group consisted of healthy individuals. * The calculated *p*-value of the statistical analysis was less than 0.05, signifying the statistical significance of the outcomes.

**Figure 7 microorganisms-12-00116-f007:**
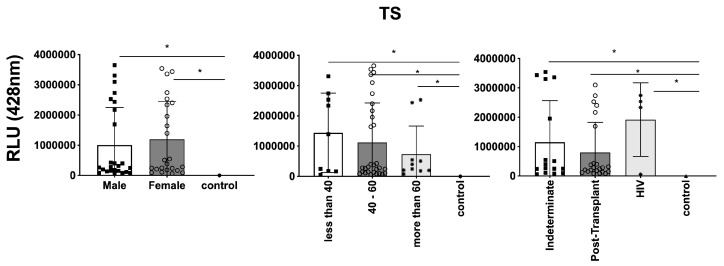
Detection of expression of TS in EVs from PCR-positive individuals with CCD and control, measured in Relative Light Units (RLUs). Categorization based on gender (**Left**), age (**Centre**), and immunosuppression status (**Right**). The “Indeterminate” group refers to immunocompetent CCD patients. The control group consisted of healthy individuals. * The calculated *p*-value of the statistical analysis was less than 0.05, signifying the statistical significance of the outcomes.

**Figure 8 microorganisms-12-00116-f008:**
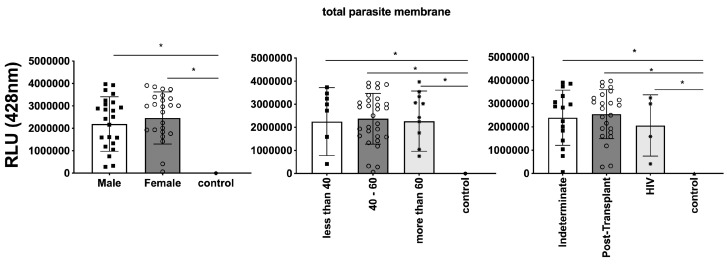
Comparison of the expression of total parasite membranes in EVs from PCR-positive individuals with CCD and control, measured in Relative Light Units (RLUs). Categorization based on gender (**Left**), age (**Centre**), and immunosuppression status (**Right**). The “Indeterminate” group refers to immuno-competent CCD patients. The control group consisted of healthy individuals. * The calculated *p*-value of the statistical analysis was less than 0.05, signifying the statistical significance of the outcomes.

**Figure 9 microorganisms-12-00116-f009:**
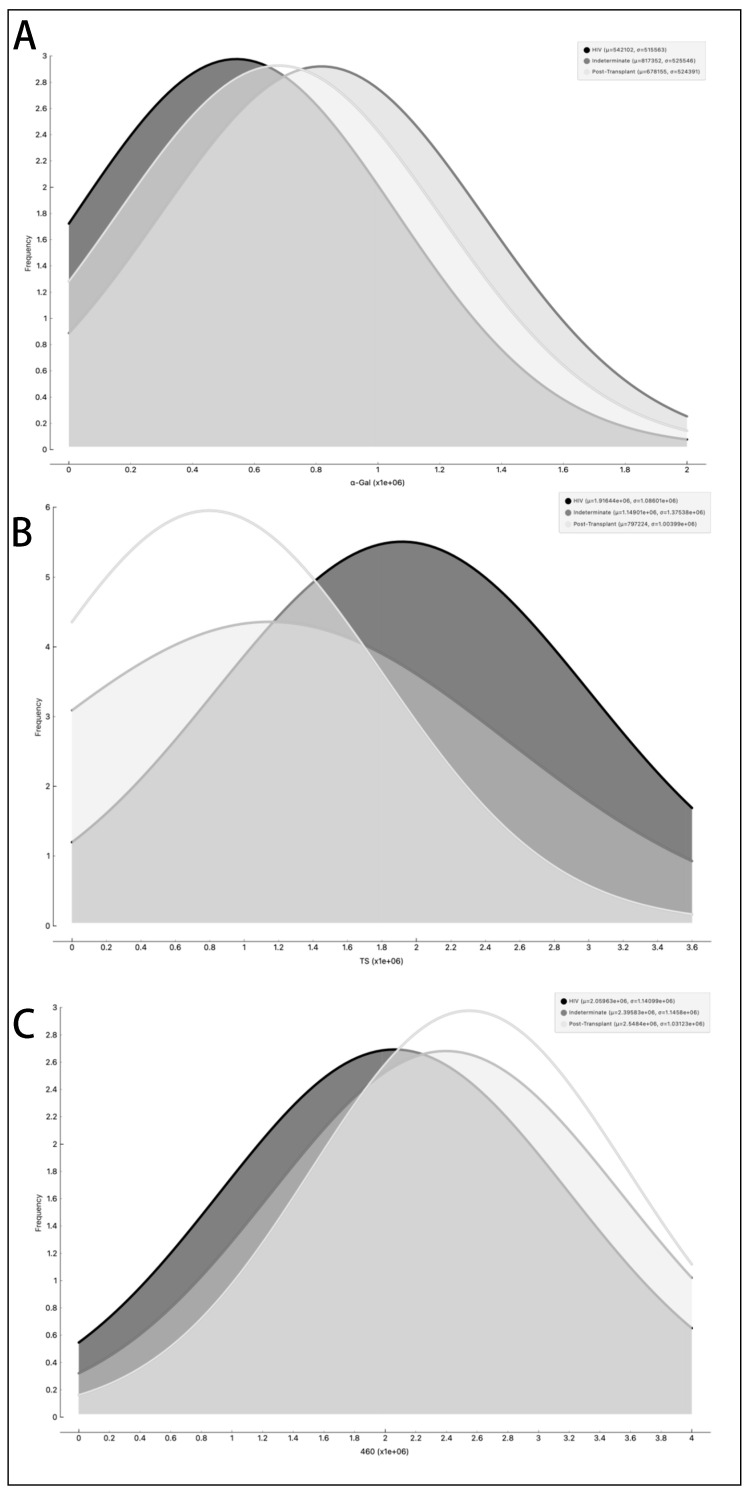
Examination of the occurrence rate, measured in Relative Light Units (RLUs), of α-Gal epitopes (**A**), trans-sialidase epitopes (**B**), and other parasite membrane epitopes (**C**) in EVs from PCR-positive individuals with CCD. The data are presented using different lines: the black line represents individuals with HIV, the dark grey line represents the Indeterminate group, and the light grey line represents post-transplant patients.

## Data Availability

The data presented in this study are available on request from the corresponding author.

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
