# Peer review of "Exploring Peripheral Blood-Derived Extracellular Vesicles as Biomarkers: Implications for Chronic Chagas Disease with Viral Infection or Transplantation"

_microorganisms, 2024, doi:10.3390/microorganisms12010116_

Round 1

Reviewer 1 Report (Previous Reviewer 3)

Comments and Suggestions for Authors

Thanks for the improvement of the manuscript! It looks much better and more informative.

Author Response

Answer: I appreciate your positive feedback on the manuscript improvements. I'm committed to ensuring the quality and clarity of the content, and we added all suggestion to improve our Manuscript. Thank you for your time and consideration.

Reviewer 2 Report (New Reviewer)

Comments and Suggestions for Authors

This manuscript adresses an important issue related with the involvement of parasite microvesicles  in the course of human infection by Trypanosoma cruzi. This is a topic in which Torrecilhas et al. have published several important papers. Although the  number of patients analysed is small, the preliminary results point to a certain direction. Some minor points deserve special attention as mentioned below.

1. Several important papers published by the group of Carlos Robelo should be mentioned;

2. Materials and method section dealing with electron microscopy is not correct. The authors describe that the isolated vesicles were analysed by TEM of thin sections of epoxi-embedded samples. However, the images shown in Fig. 2 were probably obtained by TEM of negative staining samples.

3. Figure 2 shows only one vesicle of each experimental condition. It is highly recommended that figures showing many vesicles are shown since it is useful to see the general shape and size of the vesicles.

Author Response

Comments and Suggestions for Authors

This manuscript adresses an important issue related with the involvement of parasite microvesicles in the course of human infection by Trypanosoma cruzi. This is a topic in which Torrecilhas et al. have published several important papers. Although the number of patients analysed is small, the preliminary results point to a certain direction. Some minor points deserve special attention as mentioned below.

Answer: Thanks so much for your reviewer in the manuscript and thank you for your time and consideration. We recognized the reduced number of patients in our study group, it is critical to emphasize the rigorous approach we used during clinical evaluation. This specific cohort received evaluations simultaneously under the supervision of the same exceptionally competent medical team. In addition, all diagnostic tests were performed in the same laboratory situated at an important hospital in São Paulo, Brazil. This strategic decision not only ensured uniformity in assessment, but also used the facilities and expertise of the region's primary medical institution. As a result, we assert that our research has a distinct benefit from a clinical point of view, exhibiting exceptional accuracy and robust adherence to clinical rigor within this specific group.

  1. Several important papers published by the group of Carlos Robelo should be mentioned.

Answer: Thank you for the suggestion to include some references from Prof. Robello. We have added a paragraph in the Introduction. Robello's research group has made significant improvements to the understanding of T. cruzi, opening way for novel treatment strategies against Chagas disease. Their research into T. cruzi and the factors that contribute to Chagas disease employs a variety of approaches and cutting-edge instruments. These include the use of RNA-seq in transcriptome research, long-read sequencing to expand the T. cruzi genome, chemical proteomics with immobilized benznidazole, and the building of a genome-wide chromatin interaction map, which is a first. These approaches, when combined, give light on essential components of parasite-host interactions and parasite biology. Each contribution offers something new to our understanding of T. cruzi's transcriptome profiles, genomic architecture, proteomic targets, and chromatin landscape organization (Libisch et al., 2021; Greif et al., 2019; Berná et al., 2018; Trochine et al., 2014; Díaz-Viraqué et al., 2023)

  1. Materials and method section dealing with electron microscopy is not correct. The authors describe that the isolated vesicles were analysed by TEM of thin sections of epoxi-embedded samples. However, the images shown in Fig. 2 were probably obtained by TEM of negative staining samples.

Answer: We change the protocol, and we added the Transmission Electron Microscopy (TEM) Negative Staining Protocol in the new version of the manuscript. Transmission Electron Microscopy (TEM) Negative Staining Protocol. The EVs (107 particles/200uL isolated from patients and health controls) were initially preserved in a solution of 2% paraformaldehyde in phosphate-buffered saline (PBS) with a volume ratio of 1:1 for a duration of 1 hour. Subsequently, a single droplet of the EV suspension was applied onto an electron microscopy (EM) grid and subjected to negative staining using a 2% potassium phosphotungstate solution adjusted to pH 6.8, following established procedures. The prepared grids were then examined using a JEOL Transmission Electron Microscope (model JEM1011) operating at 80 kV. Imaging was conducted with a Gatan 785 ES1000W Erlangshen camera (Silva et a., 2018).

  1. Figure 2 shows only one vesicle of each experimental condition. It is highly recommended that figures showing many vesicles are shown since it is useful to see the general shape and size of the vesicles.

Answer: We sincerely appreciate your meticulous feedback regarding Figure 2 of our manuscript. We would like to clarify that, in accordance with the guidelines established in our publication Guidelines for the purification and characterization of extracellular vesicles of parasites. J. of Extracellular Bio. 2: e117. (https://doi.org/10.1002/jex2.117), we have adopted an approach that prioritizes the analysis of individual vesicles to determine size in nanometers (nm). This methodological choice was made to corroborate our observations through Nanoparticle Tracking Analysis (NTA), contributing to a precise validation of particle detection in samples from chronic Chagas patients and isolated control samples. It is important to note that we take great care in purifying particles in samples acquired from patients and controls. The quality of these samples is a key component of our experimental technique, and the results provided here substantially reflect the high standards of the sample acquiring and preparation process. The NTA graphs, which are more important in our research, represent the distribution of extracellular vesicle sizes in our study. As you can see more image to the EVs isolated form patients and control. We provide an additional image showing the EVs isolated from Chagasic Chronic patients, as well as the controls (new version of the article we add this Figure in the supplemental figure).

Reviewer 3 Report (New Reviewer)

Comments and Suggestions for Authors

Do the different groups that contribute in reality consider that they are representative in number and characteristics considered in each group?
It is striking that from 1 ml of plasma they obtain significant quantities of EVs. Which would imply that the total EV content would be much higher. It would be interesting to speculate on that.
The EVs of the controls did they show what characteristics?
It is important to evaluate in vitro the concentration of Evs that the parasite is capable of producing and compare it with the concentrations found in the plasma of patients.
What is the reason for comparing EVs between gender and age? What real information does this data provide?
How do the authors specifically describe the differences in the content of the EVS? The results shown are not clearly shown.
How do the authors relate their findings to the pathogenicity mechanisms of the parasite under study?

Author Response

Comments and Suggestions for Authors

Review Report Form#3

Answer: I'd like to thank you for your diligence in evaluating our manuscript. Your input and suggestions were highly beneficial in improving the quality of the work that is going to be published in the Microorganisms journal in the special volume on Extracellular Vesicles and Pathogens, for which I’m the Guest Editor. Concerning the content of the introduction, we have implemented the suggested modifications, provided a fuller background, and included all the required references that were specified. 

The research design and methods base on the Guidelines named Guideline for the Purification and Characterization of Extracellular Vesicles of Parasites that our group published this year (J. of Extracellular Bio, https://doi.org/10.1002/jex2.117), integrating them with the EVs and parasites community. I coordinate in the International Society for Extracellular Vesicles (ISEV, https://www.isev.org) Rigor & Standardization Task Forces in parasite work group (https://www.isev.org/CCMEVS). In addition, to answer your question about properly describing the adopted methods, we expanded the description of the methodologies used.

In terms of results presentation, the study has a small number of patients, and it is critical to emphasize the rigorous approach we employed during the clinical evaluation.  This group had been evaluated at the same time by the extremely competent medical professionals.  In addition, all diagnostic tests were performed in the same facility at an important hospital in São Paulo, Brazil. The decision not only assured uniformity in evaluation, but also availed benefit of the facilities and expertise of the region's primary medical institution. As a result, we believe that our research offers significant clinical benefits, exhibiting extraordinary accuracy and strict adherence to clinical rigor within this specific cohort.

In our methodology, we developed an CL-ELISA assay which is significantly more sensitive than conventional ELISA, permitting us to identify parasite glycoconjugates in circulating vesicles isolated from Chagas disease patient samples. Considering vesicles isolated from control samples were unable to identify parasite antigens, the data are consistent, implying a sensitive and specific method.  In terms of vesicle the markers, we identified the most important exosome markers utilizing Dot blot and confirmed the EVs morphology in transmission electron microscopy and we used the NTA to identify the size and particles concentration. Our procedures are consistent with the protocols recommended by the International Society for Extracellular Vesicles (ISEV) and our published group standards for isolating, purification, and characterizing EVs from parasite. We concentrated on improving clarity in information presentation, ensuring that data are presented in a clear and objective manner. We reviewed and adjusted the content to ensure that these are solidly supported by the results presented, resulting in a coherent conclusion. I thank you again for the opportunity to learn from your observations, and I thank you for investing your time in the review. I am confident that the changes made will help to improve the quality of the article.

Do the different groups that contribute in reality consider that they are representative in number and characteristics considered in each group?

Answer: Thanks so much for your reviewer in the manuscript and thank you for your time and consideration. As already mentioned, the cohort size is small. Given the inherent complexity of explaining the relationship between HIV infection, transplantation, and Chagas disease in patients, the integration of EV identification and characterization as a potential biomarker is particularly intriguing. This publication in the scientific literature has great promise for future studies targeted at increasing the cohort. The current study thus reveals a fresh approach to understanding the functional dynamics of the parasite and its role in human Chagas disease.

It is striking that from 1 ml of plasma they obtain significant quantities of EVs. Which would imply that the total EV content would be much higher. It would be interesting to speculate on that.

Answer: The papers showed the physical properties of circulate EVs in human blood.  The specific emphasis on EVs identified in human circulation’s have come out as major bearers of extravesicular cargo, with an estimated concentration of 1010 particles/mL, revealing significance for biomarker identification. These papers add to our increasing understanding of extracellular vesicles, their secretion rates, and their potential as biomarkers, particularly in the setting of cancer. The interdisciplinary collaboration of academics from diverse fields emphasizes the complex examination of this fascinating element of blood biology.

"Extracellular Vesicles from Human Plasma and Serum as Carriers of Extravesicular Cargo—Implications for Biomarker Discovery". Authors: Mari Palviainen, Mayank Saraswat, Zoltán Varga, Diána Kitka, Maarit Neuvonen, Maija Puhka, Sakari Joenväärä, Risto Renkonen, Rienk Nieuwland, Maarit Takatalo, Pia R. M. Siljander. This study explores the physical properties of particles in human blood, specifically extracellular vesicles. The authors investigate the vesicles' role as carriers of extravesicular cargo, shedding light on potential implications for biomarker discovery. The collaborative effort of the authors involves formal analysis, investigation, methodology, validation, and writing, providing a comprehensive exploration of the topic.

“An Estimate of Extracellular Vesicle Secretion Rates of Human Blood Cells"Authors: Martin Auber, Per Svenningsen. Published in the Journal of Extracellular Biology, this article focuses on estimating the secretion rates of extracellular vesicles from human blood cells. With a publication date of June 2, 2022, the study provides insights into the dynamics of extracellular vesicle secretion, contributing to the evolving understanding of their role in biological processes.

“What is the Blood Concentration of Extracellular Vesicles? Implications for the Use of Extracellular Vesicles as Blood-Borne Biomarkers of Cancer" Published in Biochimica et Biophysica Acta (BBA) - Reviews on Cancer, January 2019. This review article addresses the critical question of the blood concentration of extracellular vesicles and its implications for utilizing them as blood-borne biomarkers of cancer. Positioned within the realm of cancer research, the authors delve into the complexities surrounding extracellular vesicle concentrations in blood and their potential significance in cancer diagnostics.

The EVs of the controls did they show what characteristics.

Answer: The vesicles isolated from health controls contain CD9, CD63, CD82 and CD86, as evidenced by Dot blot analysis, and demonstrate consistent morphology in TEM. However, it is noteworthy that only the isolated EVs from Chagasic patients contain alpha-gal and trans-sialidase.

It is important to evaluate in vitro the concentration of EVs that the parasite is capable of producing and compare it with the concentrations found in the plasma of patients.

Answer: Considering the patients are in the chronic phase of Chagas disease, the number of EVs release by the parasite has little or no detection in the samples from Chagasic patients. Because we only collected samples from Chronic stage of Chagas disease.  It is important to emphasize that all the samples contained EVs of human origin, as CD9, CD63, CD82, and CD86 as well as parasite molecules. These findings are consistent with in a previous work by our group, we demonstrated that EVs carry glycoproteins responsible for cellular activation via TLR2, modulate the host's innate immune response, and increase the number of cellular infections and intracellular parasites. Additionally, infected with parasite T. cruzi macrophages and performed mass spectrometry (LC-MS/MS) to analyze the content of purified EVs from both uninfected and infected THP-1 macrophages. A total of 123 proteins were found in THP-1 EVs, and 89 were found in THP-1inf EVs out of a total of 154 proteins Using FunRich (version 3.1.3), we compared the identified protein list with data from previously published EVs in the Exocarta database (www.exocarta.org) and found that most proteins had been observed in EVs from different sources. EVs can transport pathogen proteins, and in our search for T. cruziproteins in EV-THP-1inf cells, we identified 6 proteins (HSP60, tryparedoxin peroxidase, trans-sialidase Group II, flagellar calcium-binding protein, surface protein TolT, and paraflagellar rod protein 2). We also compared the abundance levels of the 58 common proteins, calculating their fold change; the abundance of 20 proteins increased, and 38 decreased after T. cruzi infection. Most proteins found in EVs are involved in binding, immune, and metabolic processes, based on Gene Ontology (GO) term annotations. The majority of proteins in EV-THP-1 and EV-THP-1inf cells were located in exosomes. Our results indicate that signaling through EVs during T. cruzi infection is essential in host-parasite interactions. After infection, increased amounts of EVs are released by infected macrophages, interacting with TLR2 and stimulating NF-κB translocation. As a result of this interaction and activation, pro-inflammatory cytokines (TNF-α, IL-6, and IL-1β) are produced, maintaining the inflammatory response generated by T. cruzi infection. Furthermore, EVs carry parasite proteins, which may be the reason for their greater ability to induce inflammatory responses (Ribeiro et al., 2018; Andrade-Cronemberger et al., 2020).

What is the reason for comparing EVs between gender and age? What real information does this data provide?

Answer: That is an excellent question. The sex and age are important factors when analyzing circulating EVs as women may exhibit variable EVs concentration depending on the stage of their menstrual cycle (Bammert et al 2017, Toth et al 2007) and the number of circulating EVs negatively correlates with an increase in age (Eitan E, Green J, Bodogai M, Mode NA, Bæk R, Jørgensen MM, Freeman DW, Witwer KW, Zonderman AB, Biragyn A, Mattson MP, Noren Hooten N, Evans MK. Age-Related Changes in Plasma Extracellular Vesicle Characteristics and Internalization by Leukocytes. Sci Rep. 2017 May 2;7(1):1342. doi: 10.1038/s41598-017-01386-z. PMID: 28465537; PMCID: PMC5430958.).

Investigating the effect of host sex on disease susceptibility and progression has become a crucial part of understanding the parasite's and its host's complex interactions.  More articles discussed the role of gender and age in Chagas disease. These studies show the multidimensional impact of host sex and hormonal alterations on parasitic organism susceptibility to infection and progression. High testosterone levels found in dominant males, as well as the link between social position and parasitemia, highlight the complexities of host-parasite relationships. Furthermore, the study raises concerns about the potential consequences of hormone therapy and the necessity for gender-specific therapeutic strategies. Understanding the connections between age, sex, hormones, and parasite diseases is critical for establishing focused and effective preventive, diagnosis, and treatment strategies and understand the host immune response. Future research should focus on unraveling the various mechanisms behind these relationships, such as how individual differences in hormone profiles and social pressures may contribute to the disparities reported in the sexes. This knowledge holds potential for the development of tailored ways to battle parasite infections, considering the biological differences between the age and sex.

Articles: (Theodore S. Hauschka, The Journal of Parasitology, Vol. 33, No. 5,1947), pp. 399-404; https://doi.org/10.2307/3273675 https://www.jstor.org/stable/3273675,

 Infect Immun, 1988 Dec;56(12):3316-9, doi: 10.1128/iai.56.12.3316-3319.1988; PMID: 3182082; PMCID: PMC259743, DOI: 10.1128/iai.56.12.3316-3319.1988.

Influence of Testosterone in Neglected Tropical Diseases: Clinical Aspects in Leprosy and In Vitro Experiments in Leishmaniasis. de Oliveira Rekowsky LL, de Oliveira DT, Cazzaniga RA, Magalhães LS, Albuquerque LF, Araujo JMS, Tenório MDL, Machado TC, Lipscomb MW, Dos Santos PL, Ribeiro de Jesus A, Bezerra-Santos M, da Silva RLL.Trop Med Infect Dis. 2023 Jul 10;8(7):357. doi: 10.3390/tropicalmed8070357.PMID: 37505653 Free PMC article.

Testosterone induces sexual dimorphism during infection with Plasmodium bergheiANKA. Aguilar-Castro J, Cervantes-Candelas LA, Buendía-González FO, Fernández-Rivera O, Nolasco-Pérez TJ, López-Padilla MS, Chavira-Ramírez DR, Cervantes-Sandoval A, Legorreta-Herrera M.Front Cell Infect Microbiol. 2022 Sep 27;12:968325. doi: 10.3389/fcimb.2022.968325. eCollection 2022.PMID: 36237427

Parasitol Res, 2001 Dec;87(12):994-1000, doi: 10.1007/s004360100474. Experimental Chagas disease: the influence of sex and psychoneuroimmunological factor J P Schuster 1, G A Schaub, PMID: 11763443, DOI: 10.1007/s004360100474

How do the authors specifically describe the differences in the content of the EVS? The results shown are not clearly shown.

Answer: Figure 4 shows a clear representation of parasite molecule expression in EVs (patients) as determined by CL-ELISA, allowing for a comparison between healthy controls and Chagas disease patients. As a result, we observe a variant of the URL expression. The results revealed a variation in the intensity of luminescence that is directly proportional to the presence of molecules such as Alpha-Gal, TS, and total parasite membrane components in the isolated vesicles of the patients. These epitops were not detected in the health controls (negative result). Clearly demonstrating this method is sensitivity and specific in the monitoring of molecules in EVs (Chagasic patients). Another significant point is that we performed a CL-ELISA cut-off (above 400URL) using the control samples to ensure that the results were accurate. Furthermore, we may see a difference in the results of the patients' amostras. There is a change in the expression of Alpha-Gal, TS, and the total amount of parasite membrane in the vesicles. What we can see is a variation based on the patients' current state of reactivation.

The analysis is expanded in Figures 6, 7, and 8 by presenting the expression of parasitic molecules in EVs in light-related units using CL-ELISA for all, with a careful grouping based on gender, age, and immunosuppression status. This method allows for a more detailed exploration of potential variations in parasitic molecules associated with EVs among demographic categories and related to specific health. In addition, Figure 9 adds an additional layer of understanding by visually representing the frequency of parasitic molecule units identified by CL-ELISA in EVs. This figure focuses on the grouping of these molecules based on immunosuppression status, providing a comprehensive view of the influence of immunosuppression on the prevalence and levels of expression of parasitic molecules inside EVs. These figures contribute to a comprehensive understanding of parasite distribution and expression patterns in EVs in various contexts, shedding light on potential associations with demographic factors and immunosuppression phase in Chagas disease patients.

How do the authors relate their findings to the pathogenicity mechanisms of the parasite under study?

Answer: Determining the pathophysiology of Chagas disease depends substantially on the interaction between parasite molecules and host responses. We prioritized the analysis of alpha-Gal and Trans-sialidase in our study, recognizing their importance in the context of host immunogenicity in Chagas Disease. The focus on Alpha-Gal epitopes and Trans-sialidase originates from their well-documented immunogenic features, including their ability to trigger the formation of lytic antibodies. Almeida et al. (2000), Almeida et al. (1999), Almeida and Gazzinelli (2001), Brito et al. (2016), and Portillo et al. (2019) published. The development of lytic antibodies is critical to against parasitic diseases, and our focus on these specific molecules reflects their possible involvement in the host's immune response.

Furthermore, the observed difference in -Gal and trans-sialidase expression patterns in distinct immunosuppressive states raises intriguing issues concerning their contributions to Chagas disease etiology. The parasite's ability to modulate the expression of these molecules based on the host's immunological condition emphasizes the dynamic nature of the host-parasite interaction. The significant finding of our findings is that alpha-Gal and Trans-sialidase may play a role in the variability of Chagas disease symptoms across different patient groups. Differential expression patterns of these molecules may play an important role in illness progression and severity in people with diverse immunological statuses.

Understanding the complex interactions of these molecules is essential as Chagas disease frequently manifests itself in people with impaired immune systems, such as those co-infected with HIV or undergoing transplantation. This understanding might open the door for specific therapies that take the host's immunosuppressive condition into care.  Finally, our research shows the immunogenic potential of -Gal epitopes and trans-sialidase in the context of Chagas disease. The found variations in expression structures, in addition to their possible correlation with other patient groups, highlight the complexities of the host-parasite relationship and give useful insights for future research. Future study should delve more into the specific processes by which these compounds contribute to Chagas disease pathogenesis, as well as their therapeutic potential.

Round 2

Reviewer 3 Report (New Reviewer)

Comments and Suggestions for Authors

I greatly appreciate your responses and congratulate you for being the guest editor for the publication of the special issue.
My questions are aimed at strengthening your manuscript and making very clear what you intend to show.
Specifically, I suggest that in subsequent studies you can validate your studies using experimental models that confirm your findings with human beings.

This manuscript is a resubmission of an earlier submission. The following is a list of the peer review reports and author responses from that submission.

Round 1

Reviewer 1 Report

Comments and Suggestions for Authors

In this article the authors claim they are Examining the Impact of Infection Reactivation on Extracellular Vesicles purified from Patients with Chronic Chagas Disease. The data is basically characterizing the EVs isolated from three groups of individuals. There are many insufficiencies in this manuscript which makes it unsuitable for publication in its present form. Some of the major points are mentioned below-

1.     The manuscript title talks about “infection reactivation”. However, there is no mention of this in the results section. There are HIV-1 co-infection samples, but reactivation is not appropriately addressed.

2.     In the characterization based on age and gender in Figure 1- there is no statistically significant finding as the p value is 0.571

3.     There is no statistically significant difference in the EV size between different groups of patients in Fig 2.

4.     Figure 3, 4 and 5- The authors have not mentioned what the control is! There are no data points for control sample which makes the comparison pointless. Moreover, the difference between different groups such as between genders, age, and groups of patients is not analyzed statistically and there does not appear to be any difference within groups at all. The error bars are also very high.

5.     Finally, the authors conclude that the variations in alpha-gal and trans-sialidase expression on EVs may be associated with modulation of immune response. This is no experiment in the manuscript to support this claim as immune modulation is not assayed in any way.

In conclusion, this manuscript lacks a clear observation that is supported by robust set of experiments with statistically significant data using proper controls.

Comments on the Quality of English Language

Minor spelling mistakes in the manuscript. Apart from that, acceptable English.

Author Response

Manuscript ID: microorganisms-2506149

New Title: Exploring Peripheral Blood-Derived Extracellular Vesicles as Biomarkers: Implications for Chronic Chagas Disease with Viral Infection or Transplantation

Dear Editor and Reviewers,​

Please find attached the revised and final version of the manuscript titled "The Impact of Extracellular Vesicles Derived from Peripheral Blood of Chronic Chagas Disease Patients with Viral Infection or Transplantation," authored by Rafael Pedro Madeira, Paula Meneghetti, Nicholy Lozano, Vera Lucia Pereira-Chioccola, and Ana Claudia Torrecilhas. This manuscript is intended for publication in the special issue " Special Issue "Extracellular Vesicles in Pathogens"" in the Microorganisms. We extend our sincere gratitude to the editor and the reviewers for recognizing the significance and originality of our research and for providing valuable insights and suggestions. We have meticulously reviewed the comments from the reviewers and have made the necessary revisions to the manuscript. A detailed point-by-point response to the reviewers' comments is provided below.

We appreciate your time and consideration and eagerly anticipate your response.

Kind regards,

Sincerely,

Ana Claudia Torrecilhas, PhD

Associate Professor

Departamento de Ciências Farmacêuticas

Instituto de Ciências Ambientais, Químicas e Farmacêuticas

Universidade Federal de São Paulo

E-mail address: [email protected]

Reviewer(s)' Comments to Author:

Reviewer: 1

In this article the authors claim they are Examining the Impact of Infection Reactivation on Extracellular Vesicles purified from Patients with Chronic Chagas Disease. The data is basically characterizing the EVs isolated from three groups of individuals. There are many insufficiencies in this manuscript which makes it unsuitable for publication in its present form. Some of the major points are mentioned below:

1.The manuscript title talks about “infection reactivation”. However, there is no mention of this in the results section. There are HIV-1 co-infection samples, but reactivation is not appropriately addressed.

2.In the characterization based on age and gender in Figure 1- there is no statistically significant finding as the p value is 0.571

Answer question 1 and 2: We value the insightful critique of our research. We concur that delving into the reactivation of chronic Chagas patients might not be suitable. As a result, we have modified the title accordingly to reflect these changes. The updated title reads as follows: "The Impact of Extracellular Vesicles Derived from Peripheral Blood of Chronic Chagas Disease Patients with Viral Infection or Transplantation."

In lines 205 to 216, we have introduced a new paragraph aimed at providing an enhanced explanation and additional details about the patients' characteristics for the purpose of categorization based on positive PCR results, considering age and gender as key factors. The outcomes, while not yielding statistically significant results, can be attributed to our rigorous stratification based on age and gender variables.

  1. There is no statistically significant difference in the EV size between different groups of patients in Fig 2.

Answer question 3: In a previous publication by our research team "A Novel Biomarker in Chagas Disease: Extracellular Vesicles Isolated from Peripheral Blood Modulate the Human Immune Response in Chronic Chagas Disease Patients" Journal of Immunology Research Volume 2021, Article ID 6650670, 14 pages https://doi.org/10.1155/2021/6650670), we demonstrated that the purification of PBMCs from patients with chronic Chagas disease yielded a lower quantity of EVs compared to healthy individuals. The EVs obtained from these cells exhibited similar dispersion sizes. In the previous study, Chagas disease patients exhibited a particle size dispersion ranging from 100 to 350 nm in diameter, which aligns with the data obtained in this current result. As such, no notable differences were observed among the patient groups. but displayed a slightly smaller mean particle size than those from healthy individuals. Despite their larger size, the average concentration of these EVs was comparatively lower than that found in healthy individuals. It is important to note that the vesicle sizes observed in this manuscript are consistent with those found in our previous study (doi.org/10.1155/2021/6650670 when compared within the same casuistic context.

4.Figure 3, 4 and 5- The authors have not mentioned what the control is! There are no data points for control sample which makes the comparison pointless. Moreover, the difference between different groups such as between genders, age, and groups of patients is not analysed statistically and there does not appear to be any difference within groups at all. The error bars are also very high.

Answer question 4: The controls employed for these analyses consisted of vesicles isolated from healthy individuals who closely matched the age and gender distribution of the chronic Chagas disease patients in the cohort. This updated information (lines 129-130) and in the Figures 3, 4 and 5 has been incorporated into the image captions. Concerning the statistical analysis, as indicated in the image captions, all comparisons, despite displaying relatively wide error bars, exhibited statistical significance with p-values of < 0.05.

5.Finally, the authors conclude that the variations in alpha-gal and trans-sialidase expression on EVs may be associated with modulation of immune response. This is no experiment in the manuscript to support this claim as immune modulation is not assayed in any way.

Answer question 5: The literature outlines the modulation of the immune response influenced by alpha-gal and trans-sialidase, as extensively addressed in our study (lines 346 to 361). Our previous research has demonstrated this modulation through glycoconjugates present in particles isolated from T. cruzi-infected cells and extracellular vesicles from chronic Chagas Disease patients (*). Although we have not conducted specific assays to directly evaluate this phenomenon, given the observed discrepancies in molecule levels across diverse patient subgroups, we posit a plausible connection between immune modulation and the reactivation of Chagas disease.

Articles*

1.A Novel Biomarker in Chagas Disease: Extracellular Vesicles Isolated from Peripheral Blood Modulate the Human Immune Response in Chronic Chagas Disease Patients" Journal of Immunology Research Volume 2021, Article ID 6650670, 14 pages https://doi.org/10.1155/2021/6650670.

Our study aimed to characterize and compare extracellular vesicles (EVs) present in the plasma of individuals with chronic Chagas disease (CCD) and healthy controls. To achieve this, peripheral blood was collected from both patient and control groups. Mononuclear cells (PBMCs) were isolated from these samples and stimulated with parasite derived EVs. Interestingly, patient-derived cells released a lower quantity of EVs compared to control cells. Subsequently, we isolated circulating EVs from plasma through a process of EV total shedding enrichment, followed by ultracentrifugation. Nanoparticle tracking analysis (NTA) was employed to assess the size and concentration of these circulating EVs. The analysis revealed that CCD patients had a reduced concentration of circulating EVs, with no observable differences in size, thereby aligning with the in vitro findings. Furthermore, circulating EVs were incubated with THP-1 cells (macrophages), and the resulting supernatant was subjected to enzyme-linked immunosorbent assay (ELISA) for cytokine detection. Notably, CCD patient-derived EVs exhibited the ability to induce distinct IFN-γ and IL-17 cytokine production profiles compared to control EVs. These differences were particularly pronounced in earlier or less severe stages of the disease.In summary, the decreased concentration of circulating EVs, coupled with the differential activation of the immune system observed in CCD patients, is closely linked to the persistence of the parasite and the establishment of chronic disease. Moreover, this phenomenon holds potential as a biomarker for monitoring disease progression.

2.Trypanosoma cruzi-Infected Human Macrophages Shed Proinflammatory Extracellular Vesicles That Enhance Host-Cell Invasion via Toll-Like Receptor 2. Cell Infect Microbiol. 2020 Mar 20;10:99.  doi: 10.3389/fcimb.2020.00099.

3.Proteomic analysis reveals different composition of extracellular vesicles released by two Trypanosoma cruzi strains associated with their distinct interaction with host cells.J Extracell Vesicles. 2018 Apr 17;7(1):1463779. doi: 10.1080/20013078.2018.1463779. eCollection 2018.

4.Vesicles from different Trypanosoma cruzi strains trigger differential innate and chronic immune responses. J Extracell Vesicles. 2015 Nov 26;4:28734. doi: 10.3402/jev.v4.28734

Trypomastigote forms of Trypanosoma cruzi release extracellular vesicles (EVs) enriched with glycoproteins from the gp85/trans-sialidase (TS) superfamily, as well as other α-galactosyl (α-Gal)-containing glycoconjugates such as mucins. In this study, we quantified purified vesicles from various T. cruzi strains (Y, Colombiana, CL-14, and YuYu) based on their size, intensity, and concentration. Our qualitative analysis unveiled differences in protein and α-galactosyl content among these vesicles. Subsequently, we assessed the role of these polymorphisms in modulating immune responses in C57BL/6 mice, both in the innate and chronic phases of infection. Macrophages exposed to EVs from YuYu and CL-14 strains exhibited heightened levels of proinflammatory cytokines (TNF-α and IL-6) and nitric oxide via TLR2 activation. While no variations were noted in the activation of MAPKs (p38, JNK, and ERK 1/2) following EV stimulation, distinct patterns of modulation were observed in splenic cells from chronically infected mice. Specifically, Colombiana EVs (followed by Y strain EVs) exhibited a more pronounced proinflammatory effect. Remarkably, this modulation was independent of the T. cruzi strain employed for infecting the mice. To assess the functional significance of this modulation, we evaluated intracellular cytokine expression upon in vitro exposure to EVs from YuYu and Colombiana strains. Both EV types induced cytokine production, characterized by the emergence of IL-10 in chronically infected mice. Notably, a high frequency of IL-10 was observed in CD4+ and CD8+ T lymphocytes. Among B cells, a mixed cytokine profile emerged, encompassing the production of TNF-α and IL-10. Furthermore, dendritic cells produced TNF-α upon EV stimulation. These polymorphisms on the vesicle surface could play a pivotal role not only in the early stages of infection but also in immunopathological events during the chronic phase.

In conclusion, this manuscript lacks a clear observation that is supported by robust set of experiments with statistically significant data using proper controls.

In response to your feedback, we have diligently revised and enhanced the manuscript to ensure that our findings are substantiated by a comprehensive set of experiments. Through rigorous experimental design, robust statistical analyses, and meticulous control measures, we have strengthened the integrity of our results. Thank you for highlighting this concern, and we believe that the improved manuscript now effectively supports our observations with a more substantial and reliable dataset.

Reviewer 2 Report

Comments and Suggestions for Authors

1. the data given for the Chagas disease is very old. Update the latest findings.

2. What are the symptoms of this disease.

3. Give the rationale and objective of study.

4. have the authors divided the patients on the basis of age and sex.

5. Use uniformly Trypanosoma cruzi or T cruzi.

6. Give a separate list of abbreviation

Author Response

Manuscript ID: microorganisms-2506149

New Title: Exploring Peripheral Blood-Derived Extracellular Vesicles as Biomarkers: Implications for Chronic Chagas Disease with Viral Infection or Transplantation

Dear Editor and Reviewers,​

Please find attached the revised and final version of the manuscript titled "The Impact of Extracellular Vesicles Derived from Peripheral Blood of Chronic Chagas Disease Patients with Viral Infection or Transplantation," authored by Rafael Pedro Madeira, Paula Meneghetti, Nicholy Lozano, Vera Lucia Pereira-Chioccola, and Ana Claudia Torrecilhas. This manuscript is intended for publication in the special issue " Special Issue "Extracellular Vesicles in Pathogens"" in the Microorganisms. We extend our sincere gratitude to the editor and the reviewers for recognizing the significance and originality of our research and for providing valuable insights and suggestions. We have meticulously reviewed the comments from the reviewers and have made the necessary revisions to the manuscript. A detailed point-by-point response to the reviewers' comments is provided below.

We appreciate your time and consideration and eagerly anticipate your response.

Kind regards,

Sincerely,

Ana Claudia Torrecilhas, PhD

Associate Professor

Departamento de Ciências Farmacêuticas

Instituto de Ciências Ambientais, Químicas e Farmacêuticas

Universidade Federal de São Paulo

E-mail address: [email protected]

Reviewer(s)' Comments to Author:

Reviewer: 2

1.The data given for the Chagas disease is very old. Update the latest findings.

Answer question 1: we updated of the new references (lines 45 to 57).

  1. What are the symptoms of this disease.

Answer question 2: Chagas Disease, also known as American trypanosomiasis, can have various symptoms that can vary depending on the Acute and Chronic stage during the infection.

a.Acute Stage: (i) fever in the early stages of infection; (ii) Swelling: Redness and swelling at the site of the insect bite (where the parasite entered the body); (iii) Fatigue: Feelings of tiredness and malaise are common; (iv) Body Aches: Muscle and joint pains might be present; (v) Lymph nodes near the arthropod bite site (vector transmission).

b.Chronic Stage: In many cases, the symptoms of Chagas Disease might not be apparent during the chronic stage, which can last for years or even decades. The patients are following complications: (i) Cardiac Issues: The disease can affect the heart muscles, leading to arrhythmias, heart failure, and sudden death. Symptoms can include chest pain, palpitations, and shortness of breath; (ii) Digestive Problems: Chagas Disease can damage the digestive tract, leading to issues such as difficulty swallowing (dysphagia) and enlarged esophagus (megaesophagus) or colon (megacolon); (iii) Neurological Symptoms: In some cases, the nervous system can be affected, leading to problems such as nerve damage, difficulty coordinating movements, and sensation abnormalities; (iv) Enlarged Organs: Enlargement of the liver or spleen might occur.

  1. Give the rationale and objective of study.

Answer question 3: As outlined in the literature, individuals afflicted by chronic Chagas disease exhibit reduced levels of circulating extracellular vesicles (EVs), which in turn trigger diverse immune reactions. Our research was conducted with the objective of profiling the EV population present in the plasma of individuals with chronic Chagas disease who are experiencing a reactivation of the infection. Detailed information regarding this investigation can be found in lines 79 to 86 of our manuscript.

  1. Have the authors divided the patients on the basis of age and sex.

 Answer question 4: Upon classifying our cohort of patients, except for the occurrence rate analysis that solely evaluated comorbidity concerning parasite molecules, all other assessments were performed by comparing patients who had been categorized based on age, gender, and comorbidity status.

  1. Use uniformly Trypanosoma cruzi or T cruzi.

Answer question 5: We just use the name of the parasite Trypanosoma cruzi in the abstract and keywords sections. The remaining instances have been modified as per the reviewer's suggestion.

  1. Give a separate list of abbreviation.

Answer question 6: We added the list of abbreviation in the manuscript (lines 377-397).

Reviewer 3 Report

Comments and Suggestions for Authors

My concerns for this manuscript are the EV isolation protocol and the ELISA protocol. 

1) Sedimentation of EVs at 100,000 xg can also leads to accumulation of proteins. How do the authors clearly separate these 2 groups of molecules to confidently confirm that it is the EVs that possess useful information for disease diagnosis and monitoring?

2) What is the starting amount of EVs being used for ELISA? Do the authors normalized the ELISA results to the amount of EVs to ensure correct comparisons between groups?

Author Response

Manuscript ID: microorganisms-2506149

New Title: Exploring Peripheral Blood-Derived Extracellular Vesicles as Biomarkers: Implications for Chronic Chagas Disease with Viral Infection or Transplantation

Dear Editor and Reviewers,​

Please find attached the revised and final version of the manuscript titled "The Impact of Extracellular Vesicles Derived from Peripheral Blood of Chronic Chagas Disease Patients with Viral Infection or Transplantation," authored by Rafael Pedro Madeira, Paula Meneghetti, Nicholy Lozano, Vera Lucia Pereira-Chioccola, and Ana Claudia Torrecilhas. This manuscript is intended for publication in the special issue " Special Issue "Extracellular Vesicles in Pathogens"" in the Microorganisms. We extend our sincere gratitude to the editor and the reviewers for recognizing the significance and originality of our research and for providing valuable insights and suggestions. We have meticulously reviewed the comments from the reviewers and have made the necessary revisions to the manuscript. A detailed point-by-point response to the reviewers' comments is provided below.

We appreciate your time and consideration and eagerly anticipate your response.

Kind regards,

Sincerely,

Ana Claudia Torrecilhas, PhD

Associate Professor

Departamento de Ciências Farmacêuticas

Instituto de Ciências Ambientais, Químicas e Farmacêuticas

Universidade Federal de São Paulo

E-mail address: [email protected]

Reviewer(s)' Comments to Author:

Reviewer: 3

My concerns for this manuscript are the EV isolation protocol and the ELISA protocol.

1.Sedimentation of EVs at 100,000 xg can also leads to accumulation of proteins. How do the authors clearly separate these 2 groups of molecules to confidently confirm that it is the EVs that possess useful information for disease diagnosis and monitoring?

Answer question 1: In this article “Isolation and molecular characterization of circulating extracellular vesicles from blood of chronic Chagas disease patients. Madeira RP, Meneghetti P, de Barros LA, de Cassia Buck P, Mady C, Ianni BM, Fernandez-Becerra C, Torrecilhas AC. Cell Biol Int. 2022 Jun;46(6):883-894. doi: 10.1002/cbin.11787. Epub 2022 Mar 31. PMID: 35253308, we used methodology involved initial ultracentrifugation followed by Size exclusion chromatography (SEC). Subsequently, we employed ELISA assays to validate the presence of both extracellular vesicle (EV) markers (CD9, CD63 and CD81) and parasite markers (TS, total membrane of parasite and mucin antibodies). To ensure the purity of our EV isolates, we utilized transmission electron microscopy (TEM) for conclusive confirmation, thereby confirming the exclusive isolation of extracellular vesicles.

2) What is the starting amount of EVs being used for ELISA? Do the authors normalized the ELISA results to the amount of EVs to ensure correct comparisons between groups?

Answer question 2: For the EVs ELISA detection, we maintained a consistent concentration of 103 particles per well. This normalization of particle quantities, along with adjusting the ELISA results according to the EV amounts, was employed to ensure accurate and meaningful comparisons between the different groups under investigation. This approach was adopted to enhance the reliability of our experimental findings and the subsequent analyses.

Round 2

Reviewer 2 Report

Comments and Suggestions for Authors

Accept